# Fecal Microbial Composition and Predicted Functional Profile in Irritable Bowel Syndrome Differ between Subtypes and Geographical Locations

**DOI:** 10.3390/microorganisms11102493

**Published:** 2023-10-05

**Authors:** Jose F. Garcia-Mazcorro, Mercedes Amieva-Balmori, Arturo Triana-Romero, Bridgette Wilson, Leanne Smith, Job Reyes-Huerta, Megan Rossi, Kevin Whelan, Jose M. Remes-Troche

**Affiliations:** 1Research and Development, MNA de Mexico, San Nicolas de los Garza 66477, Mexico; 2Instituto de Investigaciones Médico Biológicas, Universidad Veracruzana, Veracruz 91700, Mexico; 3Department of Nutritional Sciences, King’s College London, London WC2R 2LS, UK

**Keywords:** irritable bowel syndrome, microbiome, gut microbiota

## Abstract

Increasing evidence suggests a microbial pathogenesis in irritable bowel syndrome (IBS) but the relationship remains elusive. Fecal DNA samples from 120 patients with IBS, 82 Mexican (IBS-C: n = 33, IBS-D: n = 24, IBS-M: n = 25) and 38 British (IBS-C: n = 6, IBS-D: n = 27, IBS-M: n = 5), were available for analysis using 16S rRNA gene sequencing. Firmicutes (mean: 82.1%), Actinobacteria (10.2%), and Bacteroidetes (4.4%) were the most abundant taxa. The analysis of all samples (n = 120), and females (n = 94) only, showed no significant differences in bacterial microbiota, but the analysis of Mexican patients (n = 82) showed several differences in key taxa (e.g., *Faecalibacterium*) among the different IBS subtypes. In IBS-D there were significantly higher Bacteroidetes in British patients (n = 27) than in Mexican patients (n = 24), suggesting unique fecal microbiota signatures within the same IBS subtype. These differences in IBS-D were also observed at lower phylogenetic levels (e.g., higher Enterobacteriaceae and *Streptococcus* in Mexican patients) and were accompanied by differences in several alpha diversity metrics. Beta diversity was not different among IBS subtypes when using all samples, but the analysis of IBS-D patients revealed consistent differences between Mexican and British patients. This study suggests that fecal microbiota is different between IBS subtypes and also within each subtype depending on geographical location.

## 1. Introduction

The digestive tract of humans and other mammals is home to billions of microorganisms known as the gut microbiota that are crucial to health. Due to their intimate symbiosis with the host, the gut microbiota plays a role in most digestive disorders, but the relationship is elusive. Irritable bowel syndrome (IBS) is a disorder of gut–brain interaction characterized by chronic and relapsing abdominal pain and altered bowel patterns, affecting millions of people worldwide [1]. Several causes of IBS have been proposed, including food intolerance, increased intestinal permeability, bacterial overgrowth, disturbed colonic gas handling, altered motor function, aberrant host immune response, environmental pollution, brain–gut interactions, and changes in the gut microbiota [2,3,4,5,6,7]. Due to the genetic makeup and unique characteristics of human populations among geographical regions, it has been suggested that studies should focus on regional and cross-cultural differences because they are more likely to improve our understanding of IBS pathophysiology [8].

The gut (mostly fecal) microbiota has been studied in IBS patients across the globe [9] but its relationship with functional and other digestive disorders is complex. IBS consists of a range of subtypes, including diarrhea-predominant (IBS-D) and constipation-predominant (IBS-C), that may themselves have different microbiota features. Females are more frequently diagnosed with IBS and clinical symptoms tend to be more severe [10], a phenomenon likely related to sex hormones [11]. Also, the differences between healthy and afflicted patients may not only relate to microbial abundance but also to microbial interactions during IBS development [12,13]. Moreover, different techniques yield different results about the gut microbiota, for example, 16S sequencing vs. 16S probe-based methodologies [14,15,16]. Finally, increasing evidence suggests the products from microbiota metabolism, not the microbiota by itself, are also related to the disorder and may facilitate IBS diagnosis [16,17,18,19,20]. The aim of this study was to characterize the fecal microbiota of patients with IBS and to investigate the effect of IBS subtype and geographical location (Mexico and United Kingdom).

## 2. Materials and Methods

### 2.1. Patients and Samples

We enrolled patients with different IBS subtypes, with predominantly constipation (IBS-C), diarrhea (IBS-D), and both (IBS-M), from Southeastern Mexico (Port of Veracruz) and the United Kingdom (London) as part of a larger study examining the effect of a dietary supplement on the gut microbiota and IBS symptoms. The enrollment period began on 1 January 2019 and ended on 31 December 2019. The diagnosis of IBS was based on Rome IV criteria [21]. Inclusion and exclusion criteria as well as the details of other study characteristics and methodology are listed in the Appendix A and in ClinicalTrials.gov (https://classic.clinicaltrials.gov/ct2/show/NCT03803319, accessed on 14 February 2023). Key eligibility criteria were age 18–65 years of age, absence of any major system comorbidity, and without exposure to antibiotics, probiotics, or prebiotics within the preceding 4 weeks. 

All patients were recruited from primary or secondary care gastroenterology clinics. Patients provided informed consent to participate and the study was approved by the Wales Research Ethics Committee 7 (Reference 18/WA/0313). Patients provided a stool sample at the baseline of the clinical trial that was used in this analysis prior to any intervention. Stools were provided within two hours of passing, and aliquots were stored at −72 °C and −80 °C until DNA extraction and sequencing.

### 2.2. DNA Extraction and 16S Sequencing

Total genomic DNA was extracted and purified from the fecal samples using the PowerLyzer PowerSoil DNA Isolation Kit (Qiagen, Hilden, Germany) at King’s College London. Fecal samples from patients in Mexico were shipped at −20 °C using an international carrier on 20 January 2020, and arrived in London on 24 January 2020, with no temperature deviations. Following extraction, DNA samples were then shipped to Molecular Research LP (Shallowater, TX, USA) for PCR and 16S rRNA gene sequencing. More details are available in the Appendix A.

### 2.3. Bioinformatics

Raw 16S sequences were processed using default parameters in QIIME2 [22] v. 2021.11. DADA2 was used for sequence quality control and feature table construction [23]. The resulting table was filtered to remove features that were present with a frequency of <20 and presence in <10 samples [22], although this approach is debatable [24]. 

Within-sample (alpha) diversity was estimated using Faith’s phylogenetic diversity that incorporates phylogenetic differences between species in the measures of biodiversity, the Shannon index that accounts for both abundance and evenness of the taxa present, Pielou’s evenness that measures relative evenness of species richness or the number of distinct features, and the Chao1 index that estimates diversity from abundant data. Beta diversity metrics included weighted and unweighted UniFrac [25], the Jaccard similarity index, and Bray–Curtis dissimilarity. Taxonomic analysis was performed using a pre-trained Naïve Bayes classifier based on the Greengenes 13_8 with 99% OTUs. 

Prediction of functional potential was performed using PICRUSt2 [26]. We also applied a supervised learning classifier to predict three metadata classes (sex, country, and IBS subtype) of unlabeled samples by learning the composition of labeled training samples, using the feature table in the q2-sample-classifier plugin in QIIME2 [27]. Higher accuracy values from this analysis indicate a higher proportion of times that test samples are assigned the correct class (in other words, better differentiation of samples). The default Random Forest Classifier method was used with either the default fraction of input samples used for classifier testing (20%) or 30% used by Manandhar et al. [28] in their study of inflammatory bowel disease. The procedure was performed for 10 independent iterations.

### 2.4. Statistical Analyses

A chi-squared test was used to compare the proportions of samples within categories (e.g., sex vs. IBS subtypes). Relative abundance data (i.e., proportions of 16S sequences) lead to spurious correlations (Appendix A) and were therefore included here only as part of routine descriptive and statistical analysis. Analysis of relative abundances or centered log-ratio (clr)-transformed data was performed in SAS University Edition with the MIXED or the Kruskal–Wallis (K-W) test in the NPAR1WAY procedure depending on the normality of residuals. Alpha diversity metrics were compared using the K-W test in the visualizer alpha-group-significance within the diversity plugin in QIIME2. The non-parametric ANOSIM and PERMANOVA, as well as PERMDISP, determined whether the variances of groups of samples were significantly different and were used to compare beta diversity metrics. STAMP [29] was used to analyze PICRUSt2 predictions. 

## 3. Results

### 3.1. Patients and Samples

A total of 120 fecal samples (n = 94 females, n = 26 males; n = 82 Mexico, n = 38 U.K.) were available for analysis. There was no significant difference in age (*p* = 0.42), Body Mass Index (BMI, *p* = 0.11), or sex percentage (*p* = 0.67, chi-squared test) among IBS subtypes. However, the proportions of patients among IBS subtypes were not equal in Mexico and the U.K. (*p* < 0.001, chi-squared test), due to the high number of patients in the U.K. with IBS-D (n = 27, 71.1%) compared with the other subtypes (Table 1). The numbers and characteristics of patients with IBS-D were similar between the two countries (Table 1).

### 3.2. Sequence Features

The original feature table (without filtering) contained 13,348 Amplicon Sequence Variants (ASVs). The filtering procedure (ASVs with a frequency of <20 and presence in <10 samples were discarded) generated a total of 379 ASVs (~3% of original features) among all samples. The filtered table contained a total of 8.5 million 16S sequences (minimum: 29,220, maximum: 161,385 16S sequences per sample).

### 3.3. Taxonomy

At the phylum level, Firmicutes (mean: 82.1%), Actinobacteria (10.2%), and Bacteroidetes (4.4%) were the most abundant taxa, followed by Proteobacteria (2.8%) and Verrucomicrobia (<0.1%, undetected in 70%, or 84/120 samples) (Table 2).

### 3.4. Analysis of Relative Abundances at the Phylum Level

As mentioned above, relative abundance (i.e., proportions of 16S sequences) data lead to spurious correlations but because this is the most common analysis and description strategy, we included these data here. Using all samples (n = 120), K-W tests showed no significant differences in relative abundance for Firmicutes (*p* = 0.79), Actinobacteria (*p* = 0.77), Bacteroidetes (*p* = 0.14), Proteobacteria (*p* = 0.55), or Verrucomicrobia (*p* = 0.69) among IBS subtypes (note that the residuals from PROC MIXED in SAS showed not normally distributed residuals for all these comparisons). Also, using all samples, no differences in relative abundance were found between genders but the U.K. samples had more Bacteroidetes (*p* = 0.02) and less Proteobacteria (*p* = 0.008) compared with the Mexican patients. Separate analyses of samples from Mexico (n = 82) or women (n = 94) also showed no significant differences in the main phyla among IBS subtypes.

### 3.5. Analysis of Clr-Transformed Data at the Phylum Level

Similar to the analysis of relative abundance, the analysis of clr-transformed data using the K-W test on all samples (n = 120) also showed no differences in the abundance of the main phyla among IBS subtypes, with the exception of a trend for lower Bacteroidetes in IBS-M compared with the other IBS subtypes (*p* = 0.08). There was also no difference between genders, but the U.K. samples had more Bacteroidetes (*p* = 0.01 K-W test, *p* = 0.02 with PROC MIXED) and less Proteobacteria (*p* = 0.004 K-W test, *p* = 0.008 with PROC MIXED) compared with the Mexican samples. Interestingly, separate analysis of patients from Mexico (n = 82) showed a significant difference in Bacteroidetes among IBS subtypes (*p* = 0.04 using both K-W test and PROC MIXED, Figure 1), with a higher abundance in IBS-C compared with both IBS-D and IBS-M (*p* = 0.03). The pattern of variation (Table 2) and the higher abundance of Bacteroidetes in patients from Mexico with IBS-C was not revealed in the analysis of relative proportions described above (*p* = 0.158 in the K-W test). Separate analysis of samples from females (n = 94) showed no difference in any phyla among IBS subtypes. Age and BMI as continuous explanatory variables were not significantly related to the variation in the clr-transformed data for all main phyla using all samples or Mexican samples only (Appendix A).

### 3.6. Analysis of Clr-Transformed Data from IBS-D at the Phylum Level

The comparison of clr-transformed data from Mexican (n = 24) and British (n = 27) patients with IBS-D revealed a significantly higher abundance of Bacteroidetes in patients in the U.K. (*p* = 0.005 with slice option = Disease_subtype in lsmeans statement in PROC MIXED; *p* = 0.02 Wilcoxon test, Figure 2). The trend for higher Bacteroidetes in the U.K. samples with IBS-D was also observed in the relative proportions (Table 2). No differences were found for the other phyla.

### 3.7. Analysis of Clr-Transformed Data at the Genus Level

At the genus level, we detected 341 taxa whilst unassigned taxa at the domain or phyla level (n = 10) were removed. We performed clr-transformation using the 110 most abundant taxa (~1/3 of original taxa, lowest: 481 sequences across all samples for *Shuttleworthia*, highest: 2,391,110 sequences for *Blautia*). Several taxa were found to be significantly different between IBS subtypes or countries while very few differences were found between genders (e.g., an unassigned taxon within the family Fusobacteriaceae) (Appendix A). Compared with patients from Mexico, those from the U.K. were characterized for having more *Faecalibacterium* (*p* < 0.001), Erysipelotrichaceae (*p* = 0.007), *Bilophila* (*p* < 0.001), and *Coprococcus* (*p* = 0.009), and less Lactobacillales (unassigned genus, *p* < 0.001), *Staphylococcus* (*p* = 0.005), *Enterococcus* (*p* < 0.001), *Lactococcus* (*p* < 0.001), *Streptococcus* (*p* = 0.002), *Catenibacterium* (*p* < 0.001), *Eubacterium* (*p* = 0.005), Enterobacteriaceae (unassigned genus, *p* < 0.001), and *Desulfovibrio* (*p* = 0.004). 

The analysis of Mexican samples (*n* = 82) revealed higher levels of *Bacteroides* in patients with IBS-C compared with IBS-M (*p* = 0.027), higher levels of an unknown genus of Rikenellaceae (Bacteroidales) in IBS-C compared with both IBS-D (*p* = 0.008) and IBS-M (*p* = 0.021), lower levels of *Streptococcus* in IBS-C compared with IBS-D (*p* = 0.026), and higher levels of Ruminococcaceae (*p* = 0.026) and *Faecalibacterium* (*p* = 0.014) in IBS-D vs IBS-M. Of all these taxa, *Streptococcus* was the only one that was correlated with age and BMI (*Streptococcus* was lower in IBS-C compared with IBS-D and this difference significantly increased with age and BMI). All other genera remained similar among IBS subtypes. Relevant taxa for gut health such as *Akkermansia* (average: 0.05%, undetected in 86/120 samples), *Bifidobacterium* (average 9.9%, undetected in 18/120 samples), and *Oscillospira* (average: 1.9%, undetected in 2/120 samples) did not differ between any subsets of samples.

### 3.8. Analysis of Clr-Transformed Data from IBS-D at the Genus Level

Significantly higher *Bacteroides* (*p* = 0.01), *Parabacteroides* (*p* = 0.02), an unassigned genus within the family Rikenellaceae (*p* = 0.05), and *Bilophila* (*p* < 0.001) were found in patients from the U.K. vs. Mexico with IBS-D. Higher Lactobacillales (unassigned genus, *p* = 0.02), Enterobacteriaceae (unassigned genus, *p* < 0.0001), *Staphylococcus* (*p* = 0.04), and *Streptococcus* (*p* < 0.001) were found in Mexicans vs U.K. patients with IBS-D. Although the differences in the number of patients for each IBS subtype prevent us from comparing IBS-C and IBS-M, it is interesting that all comparisons between IBS subtypes for Enterobacteriaceae and Enterococcus between countries showed significant differences (*p* < 0.005 for all comparisons). No differences were found for important members of the gut microbiota such as *Akkermansia*, *Bifidobacterium*, *Faecalibacterium,* and *Oscillospira*.

### 3.9. Alpha Diversity Analyses

The visualizer alpha-group-significance within the diversity plugin in QIIME2 was used to compare alpha diversity metrics with 29,000 sequences per sample. Using all samples (n = 120), there was no significant difference in the Shannon index (*p* = 0.61), evenness (*p* = 0.65), ASVs (*p* = 0.13), or the Chao1 index (*p* = 0.14) among the IBS subtypes, and a non-significant difference was recorded in Faith’s PD (*p* = 0.056 all groups, IBS-C > IBS-D with *p* = 0.02) (Figure 3). A Venn diagram also showed only a few ASVs that were unique to each subtype. 

Interestingly, none of the alpha diversity metrics were significantly different between countries, genders, ages, or BMI groups. These results were similar when using a lower number of sequences per sample (i.e., 10,000 vs. the original 29,000 sequences), suggesting that the results are not related to sequencing depth. We also analyzed the effect of age and BMI as continuous explanatory variables with either sex or country as independent variables in SAS (Appendix A). The most significant result of this analysis suggests that BMI is a variable to look for when looking at differences between countries but not between genders. Additional analyses of Mexican patients only (n = 82) showed no differences among IBS subtypes in any alpha diversity metrics (which is interesting given the significant abundance of several taxa described above), while the analysis of all females (n = 94) also showed no differences with the exception of Faith’s PD that were significantly higher in IBS-C compared with IBS-D (*p* = 0.03 K-W pairwise), a result that was also suggested in the analysis of all samples. Finally, the comparison of metrics between patients with IBS-D (similar number of samples between Mexico, n = 24, and U.K., n = 27), revealed a higher Faith’s PD in Mexican patients compared with the U.K. (*p* = 0.04), a trend for a higher number in ASVs (*p* = 0.07) and Chao1 indices (*p* = 0.06), but not in the Shannon or evenness results. Overall, the results suggest that IBS subtypes are not related to differences in alpha diversity metrics and that these results are not related to age, BMI, gender, or country of origin. However, as described above, patients from different geographical areas with the same IBS subtype (in this case, IBS-D) may display differences in alpha diversity metrics.

### 3.10. Beta Diversity Analyses

There was little or no difference in weighted or unweighted UniFrac, Jaccard, or Bray–Curtis distances among IBS subtypes according to all tests, while most comparisons between countries produced statistical significance (Figure 4, Table 3 and Table 4).

Additional analyses of distances from IBS-D patients between Mexican samples, U.K. samples, and Mexico–U.K. samples, showed significant differences in all beta diversity parameters (weighted UniFrac: *p* = 0.0482; unweighted UniFrac: *p* < 0.0001; Jaccard: *p* < 0.0001; Bray–Curtis: *p* = 0.005) according to K-W tests in SAS. The difference in results of unweighted and weighted UniFrac tests suggests that the differences between patients from Mexico and the U.K. with IBS-D were not related to the microbial numbers or abundance but rather to the types of microbes present in the samples.

### 3.11. Prediction of Functional Potential

The analysis of PICRUSt2 data in STAMP showed a total of 112 significant features (Kruskal–Wallis H-test, adjusted *p* < 0.01) using the variable country with two levels, and 73 features when using the variable disease*country, with six levels (Mexico and U.K. with all three IBS subtypes). In our experience, it is uncommon to find such a number of significant features, particularly with the stringent adjustment in STAMP. No significant features were found between genders, BMI, or age groups, with three levels, or among IBS subtypes. It is interesting that 7 out of 20 most significant features (lowest *p* values) that were different between countries were involved in menaquinones and demethylmenaquinones biosynthesis, and all of those 20 features were higher in Mexican patients. Also, 19 of those 20 features were also detected as significantly different when using the variable disease*country, with six levels (Mexico and U.K. for all IBS subtypes) (Appendix A). In addition to the 20 most significant features, we also included the methanogenesis pathway for the involvement of methane in IBS, particularly the constipation-predominant form [30], and found that within the IBS-D subgroup, patients from Mexico had a lower relative abundance of genes related to that pathway (*p* = 0.0003). No significant differences were found between IBS subtypes, BMI, or age groups when using data from Mexicans only. Overall, these results confirm the differences between countries and add support to the need for investigating regional differences to obtain a better understanding of the role of the microbiota in IBS pathophysiology. 

### 3.12. Supervised Learning Classification

The area under the curve (AUC) is a measure of performance with higher values representing better accuracy of the classification procedure. AUC values for all three IBS subtypes ranged from 0.33 to 0.78 (overall average: 0.58), with IBS-M showing lower average values (IBS-C: 0.59; IBS-D: 0.63; IBS-M: 0.51). Average AUC values were identical for females and males (0.55) and for Mexico and U.K. (0.97). The comparison of AUC values between 20% and 30% of input samples used for classifier testing did not show statistical differences for any class variable. Overall, these results indicate that the class variable country was the most accurate at predicting microbial composition.

## 4. Discussion

The gut microbiota is related to health and disease but its relationship with IBS and other GI functional disorders remains elusive. This may not be related to a lack of biological relationship between the two, but perhaps to high inter-individual differences and also to the biological uncertainty associated with the clinical classification of IBS subtypes [31]. To the best of our knowledge, this is the largest study of the fecal microbiota associated with patients diagnosed with IBS from Mexico, and one of the most comprehensive views of IBS-associated microbiota from two geographical regions within the same time frame using the same clinical criteria, analysis strategies, and analytical runs. 

Most papers on the gut microbiota present results about microbial abundances but the real abundances of the different taxa are difficult to estimate using 16S rRNA gene sequencing, in part because they depend not only on their true numbers in nature but also on the DNA extraction method [32], sample processing [33], analysis strategy [34], luminal density [35], and many other unknowns [36]. One meta-analysis of hundreds of human fecal samples from several countries showed that both Firmicutes and Bacteroidetes varied evenly from ~25% to ~70% [37], which heavily contrasts with the reported relative abundances in this investigation (~80% Firmicutes and ~4% Bacteroidetes). A recent paper from our research team that used another DNA extraction method also reported high levels of Firmicutes in human fecal samples (average: 83% from 26 samples) with very low levels of Bacteroidetes (~1%) from patients with celiac disease and non-celiac gluten sensitivity as well as healthy controls from the same region of Mexico (Port of Veracruz) [38]. However, another more recent paper on the fecal microbiota from constipated patients from the same geographical area showed much lower levels with an average of 56% for Firmicutes and 16% for Bacteroidetes using that same DNA extraction strategy [39]. The wide variation may reflect the extensive variation among individuals in that region. Regardless, in this study, all samples were treated in the same way and can therefore be meaningfully compared, and the lack of significant differences in the main taxa does not necessarily rule out a relationship between the gut microbiota and IBS. In fact, one recent study suggested that the metabolites are the main drivers of differences between IBS subtypes [16]. This is not a trivial matter, particularly when considering the fact that members of the “same” taxon (e.g., Bacteroides), based on 16S similarities, can behave very differently in nature, something that has been demonstrated in other environments [40]. This important issue has not been well studied in studies of the gut microbiome. 

*Streptococcus* is an interesting taxon (mean 2.2% of all 16S reads across all samples) in the context of IBS [41] and obesity [42] that was shown to be higher among Mexicans, increased in IBS-D patients from Mexico vs. U.K., and decreased in patients from Mexico with IBS-C compared with IBS-D. Moreover, from all taxa that were different among IBS subtypes in Mexican samples, *Streptococcus* was the only one correlated with age and BMI. Although the relationship between *Streptococcus* and methane metabolism is not well studied, the finding of lower *Streptococcus* in Mexican samples with IBS-C is interesting in the context of the findings presented by Aja-Cadena et al. [43] that used a patient population from the same area and showed that IBS-C patients had a greater methanogenic propensity. This is in line with previous findings [44]. Another study reported that the abundance of *Streptococcus* in the mucosa-associated microbiota is relatively high in Japanese patients with IBS-C [45]. Later, Xia et al. [46] suggested a synergistic pattern of neurotransmitter metabolism between *Streptococcus* and *Shigella*, a member of the family Enterobacteriaceae, in patients with IBS-D. An exploration of gutMGene [47] showed that members of this taxon are negatively correlated with the expression of several chemokine genes, some of which have reported antimicrobial properties, a phenomenon that may also partly explain differences in *Streptococcus* abundance. *Streptococcus* deserves attention in IBS and perhaps other digestive ailments. 

This study shows that IBS subtypes do not have differences in alpha diversity metrics. There was also no difference according to age, BMI, gender, or country, and the analysis of patients from Mexico only showed marginal differences. In our experience, this was unexpected because differences in microbial abundances are usually accompanied by differences in within-sample alpha diversity. Like our results, Pozuelo et al. [17] used the Chao1 index to estimate alpha diversity in fecal samples and showed no differences among IBS subtypes (n = 113) and no change over a one-month sampling period. In that study, healthy controls had higher Chao1 indices but individual comparisons between healthy controls and each of the IBS subtypes showed that only IBS-D had a trend towards a lower diversity. Another large study (n = 490 with IBS) showed that healthy controls had higher Shannon indices, as well as it being higher in IBS-C (n = 185) compared with IBS-D (n = 86) [48], but no other alpha diversity metric was used to confirm this finding. More recently, Su et al. [7] used sequence variants present in more than 50 samples with a total relative abundance higher than 0.01%, and showed that subjects with IBS-D or IBS-U (unclassified), but not IBS-C, had significantly lower bacterial diversity compared with non-IBS controls, as assessed via the Shannon diversity index. Another study showed that probiotic administration in IBS-D patients was not related to changes in alpha and beta diversity in both the adherent and luminal microbiota, despite significant changes in several taxa of the luminal microbiota [49]. The relationship between microbial diversity and IBS deserves attention, particularly in the context of microbial ecology and low abundant rare microbes [50,51,52]. 

The specific microbial signatures between patients with IBS-D in Mexico and the U.K. were not unexpected because host genetics is a strong factor in shaping gut microbiomes [53]. Bacteroidetes were significantly higher in IBS-D patients in the U.K. and this difference also included taxa at the genus level within Bacteroidetes and other phyla. Overall, these results confirm the fact that subjects may be diagnosed with the same subtype of IBS from two countries, yet show differences that make them unique to their group. As explained by others, this geographical variation in gut microbiomes is related to the close interaction between host genetics, diet and dietary habits, and other factors [54,55]. Indeed, investigating patients from different geographical locations can improve our understanding of the pathophysiology of GI and other disorders [8]. 

Our results need to be taken cautiously in clinical practice. The gut microbiota has been called an evolving fingerprint [56] because it is always unique to each individual and at the same time shows a unique pattern of variation over time. Some researchers have pushed toward the use of probiotics and prebiotics in IBS and other GI disorders, supported by randomized studies that have shown the potential of these supplements in alleviating symptoms [57,58]. However, one study showed unfavorable effects on symptoms of a candidate probiotic based on one strain of *Lactobacillus plantarum* [59]. The authors explained this was possibly due to different probiotic properties and did not exclude a too-high dose, the latter being very unlikely. Given the large heterogeneity of IBS manifestations, particularly among geographical areas [31], and the ambiguity related to the concept of healthy microbiome and dysbiosis [60], the solution to improve the quality of life in IBS patients should go beyond probiotics and prebiotics and include a combination of dietary, exercise, and psychological [61] interventions, particularly in patients with concomitant depression [62]. 

This study has several limitations. First, the inclusion of healthy controls could have served to compare the fecal microbiota of IBS patients. Second, we only collected one single fecal sample per patient, which could have obscured over time variation [63], although one study showed stability of the fecal microbiome over a one-month period in both healthy controls and IBS patients [17]. Third, DNA extraction methodologies have a very strong effect on our view of gut microbial communities, although here we only used the same commercial kit for all samples. Fourth, feces usually do not reflect the microbiota in the mucus, e.g., *Akkermansia* spp., which are crucial for gut health and homeostasis [64]. This issue is particularly relevant due to the possible proximal “expansion” of indigenous colonic microbes in IBS [65]. Fifth, the mixture of stool material, common to most research papers on the fecal microbiota, prevents the accurate evaluation of microbial communities due to the destabilization of the biostructure [33]. This issue remains a challenge, especially in samples from patients with diarrhea. Finally, most publications have focused on bacteria, but the mycobiome deserves attention in gut disorders including IBS [66]. 

It is important to note that more patients were included in the enrollment period in Mexico (in a 2:1 ratio) than in the U.K., and that in Mexico the proportion of patients with IBS-C was higher but IBS-D predominated in the U.K. Although this may represent a selection or reference bias between countries, the difference in the numbers of patients with different IBS subtypes may not only depend on prevalence within each region but also on access to health services, ease of improving symptoms with dietary and other lifestyle changes, and socio-cultural differences.

In summary, this research suggests that different IBS subtypes may be associated with different fecal microbial populations, but this relationship remains obscure. We suggest that the inter-individual variation and the biological ambiguity—both clinically relevant—behind the classification of IBS subtypes may partly explain our results. Clear differences were found between the microbiota of patients with IBS-D in Mexico and the U.K., suggesting that individuals diagnosed with the same subtype of IBS can show unique gut microbial patterns depending on host genetics and environmental factors from their place of residence. More studies investigating gut microbial metabolites, particularly potential biomarkers, are warranted. 

## Figures and Tables

**Figure 1 microorganisms-11-02493-f001:**
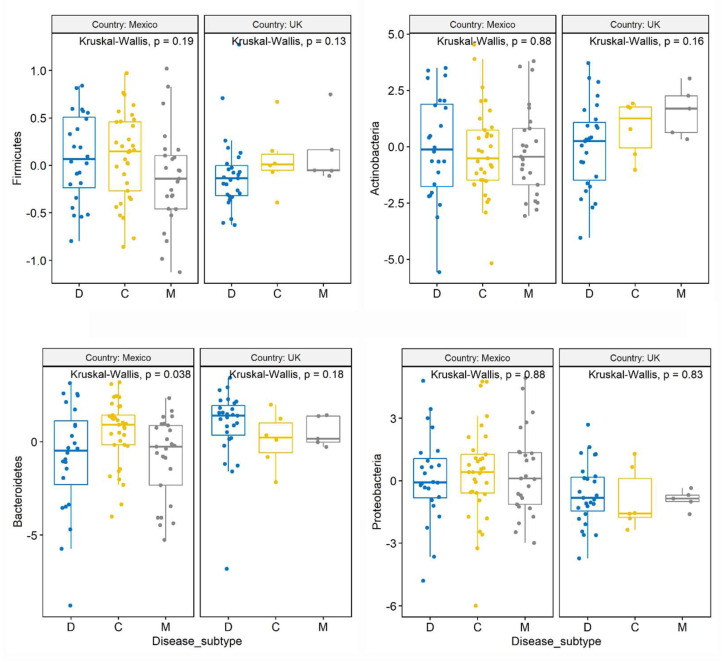
Box plots with jitter comparing clr-transformed data among IBS subtypes at the phylum level.

**Figure 2 microorganisms-11-02493-f002:**
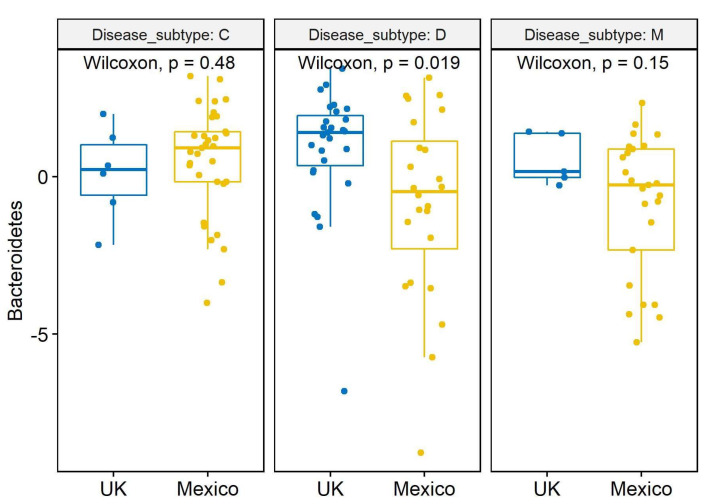
Box plots with jitter comparing clr-transformed data between countries for the phylum Bacteroidetes. The similar number of samples from samples from the U.K. (n = 27) and Mexico (n = 24) from patients with IBS-D allows a meaningful comparison.

**Figure 3 microorganisms-11-02493-f003:**
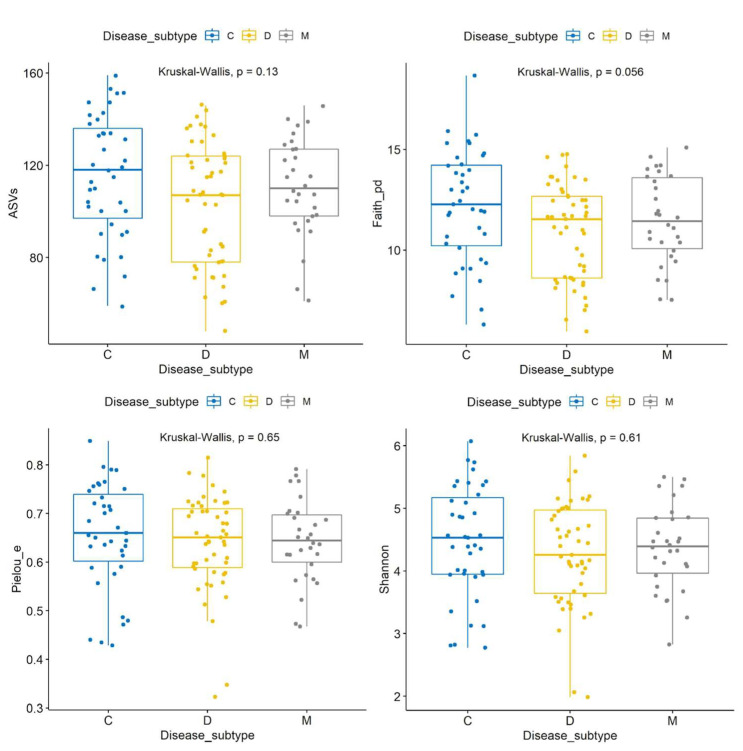
Box plots with jitter comparing alpha diversity metrics. All samples were used in this analysis (n = 120).

**Figure 4 microorganisms-11-02493-f004:**
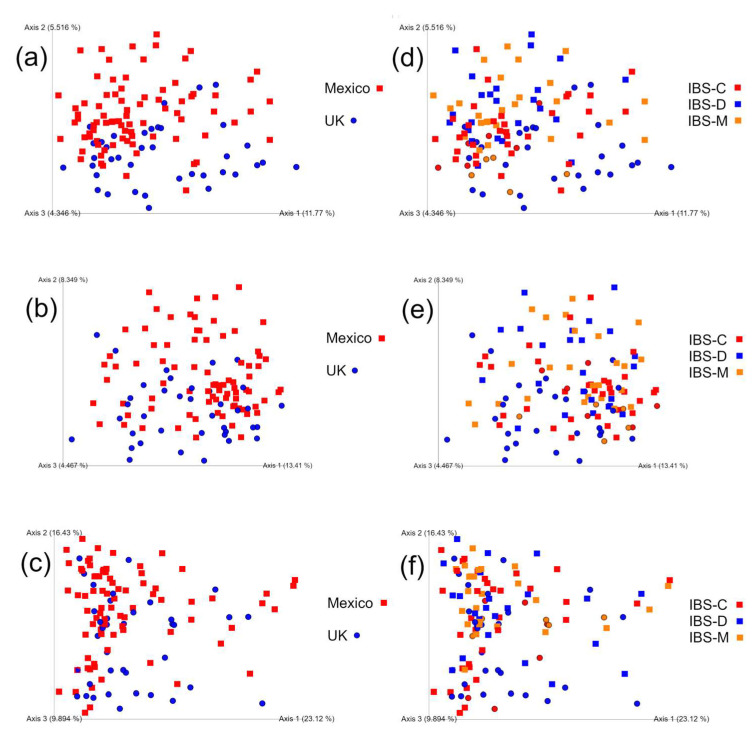
PCoA plots with Jaccard (**a**,**d**), unweighted (**b**,**e**), and weighted (**c**,**f**) UniFrac distances. The plots were separated to show samples according to country (**a**–**c**) and IBS subtypes (**d**–**f**). In all cases, samples from Mexico are squares and those from the U.K. are circles.

**Table 1 microorganisms-11-02493-t001:** Subject demographics of patients with the three IBS subtypes.

	IBS-D (n = 51)	IBS-M (n = 30)	IBS-C (n = 39)	*p* Value
Age, years, mean (SD)	32.5 (11.3)	31.1 (14.2)	31.9 (12.7)	0.42
BMI, kg/m^2^, mean (SD)	25.8 (5.0)	27.1 (4.6)	24.6 (4.3)	0.11
Gender, n (%)				
Females (n = 94)	38 (40.4)	24 (25.6)	32 (34.0)	0.67
Males (n = 26)	13 (50.0)	6 (23.1)	7 (26.9)	
Country, n (%)				
Mexico (n = 82)	24 (29.3)	25 (30.5)	33 (40.2)	<0.001
U.K. (n = 38)	27 (71.1)	5 (13.2)	6 (15.8)	

**Table 2 microorganisms-11-02493-t002:** Relative proportions ^1^ (mean ± SD) of 16S rRNA gene sequences for the most abundant taxa at the phylum level.

Phylum	IBS-D	IBS-M	IBS-C
Firmicutes	81.8 ± 12.9%	82.1 ± 15.5%	82.6 ± 12.2%
	Mexico: 81.6 ± 14.5%	Mexico: 82.3 ± 16.7%	Mexico: 82.4 ± 13.0%
	U.K.: 81.9 ± 11.8%	U.K.: 80.8 ± 8.3%	U.K.: 84.0 ± 6.6%
Actinobacteria	10.6 ± 12.5%	11.4 ± 14.8%	8.9 ± 11.7%
	Mexico: 11.0 ± 13.5%	Mexico: 10.6 ± 15.7%	Mexico: 8.5 ± 12.4%
	U.K.: 10.2 ± 11.7%	U.K.: 15.6 ± 9.2%	U.K.: 11.2 ± 6.8%
Bacteroidetes	5.2 ± 5.5%	2.7 ± 2.7%	4.7 ± 4.6%%
	Mexico: 3.7 ± 5.2%	Mexico: 2.7 ± 2.9%	Mexico: 4.9 ± 4.8%
	U.K.: 6.5 ± 5.5%	U.K.: 2.9 ± 1.7%	U.K.: 3.4 ± 3.3%
Proteobacteria	1.9 ± 4.0%	3.5 ± 6.3%	3.3 ± 6.4%
	Mexico: 2.9 ± 5.5%	Mexico: 4.1 ± 6.8%	Mexico: 3.7 ± 6.8%
	U.K.: 1.2 ± 1.6%	U.K.: 0.5 ± 0.2%	U.K.: 0.9 ± 1.2%
Verrucomicrobia	0.03 ± 0.1%	0.08 ± 0.4%	0.05 ± 0.2%
	Mexico: 0.04 ± 0.2%	Mexico: 0.1 ± 0.5%	Mexico: 0.05 ± 0.2%
	17/24 undetected	19/25 undetected	25/33 undetected
	U.K.: 0.02 ± 0.1%	U.K.: <0.01 ± 0.02%	U.K.: 0.03 ± 0.01%
	16/27 undetected	3/5 undetected	4/6 undetected

^1^ Relative abundance data (proportions of 16S sequences) lead to spurious correlations and were therefore included only as part of routine descriptive statistics.

**Table 3 microorganisms-11-02493-t003:** Summary of statistical results (p values) for the comparison of beta diversity between all three IBS subtypes.

Beta Diversity Metric	Anosim	PERMANOVA	Perm-Disp
Weighted UniFrac	*p =* 0.339	*p =* 0.033	*p =* 0.573
Unweighted UniFrac	*p =* 0.192	*p =* 0.068	*p =* 0.594
Jaccard	*p =* 0.438	*p =* 0.056	*p =* 0.603
Bray–Curtis	*p =* 0.075	*p =* 0.014	*p =* 0.655

**Table 4 microorganisms-11-02493-t004:** Summary of statistical results (p values) for the comparison of beta diversity between countries.

Beta Diversity Metric	Anosim	PERMANOVA	Perm-Disp
Weighted UniFrac	*p =* 0.114	*p =* 0.001	*p =* 0.417
Unweighted UniFrac	*p =* 0.002	*p =* 0.001	*p =* 0.900
Jaccard	*p =* 0.001	*p =* 0.001	*p =* 0.718
Bray–Curtis	*p =* 0.081	*p =* 0.001	*p =* 0.292

## Data Availability

Raw 16S sequencing results are available at the Sequence Read Archive (BioProject PRJNA633542).

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
