# Peer review of "Fecal Microbial Composition and Predicted Functional Profile in Irritable Bowel Syndrome Differ between Subtypes and Geographical Locations"

_microorganisms, 2023, doi:10.3390/microorganisms11102493_

Round 1

Reviewer 1 Report

Garcia-Mazcorro et al. analyzed the fecal microbiome from three different types of IBS (constipation, diarrhea, and mixed) patients from Veracruz, MX and London, UK. Sequencing of 16S rRNA showed more consistency in microbial flora within IBS subtypes than between geographic location supporting the hypothesis that each subtype of IBS has a unique microbiome. Overall, the study is well conducted, and the findings will be of interest to the IBS and microbiome community. The central issue is that there are many prior publications in the role of microbiota in IBS subtypes that have heterogeneity or conflicting findings. Furthermore, most of the studies are phenomenological or correlative, with no clear mechanism(s) identified. I have a few suggestions that may improve the manuscript.

1.Is it feasible to show the overlap in the OTU for each IBS subtype between the MX and UK populations? As noted by the authors diet, lifestyle, and other factors (sex, medications, age et al.) can be confounding. However, are there any species/OTU that are “core” for each subtype? A Venn diagram (or enrichment analysis) may be helpful? I am not sure that the analysis at the phylum level is sufficient for resolving the key differences. Furthermore, as the authors note we don’t know what a “healthy” microbiome is for either population.
2.As IBS presents as an activation of the immune system in the bowels, can the authors comment on the localization of any of the OTU identified? There are several tools like gutMgene ( doi: 10.1093/nar/gkab786) that can be used, perhaps. As the authors note, the fecal microbiota represents a mix of microbes along the GI tract.
3.Can the authors include a 3x6 matrix (for anyone, e.g., Unifrac, Bray-Curtis) of the beta-diversity measures for 3 IBS subtypes and the two countries?

Author Response

Dear reviewer,

Thank you for your time. Here you can find the response to your queries and suggestions. Also, please see the attachment to find our response to all reviewers.

********

Reviewer 1

Garcia-Mazcorro et al. analyzed the fecal microbiome from three different types of IBS (constipation, diarrhea, and mixed) patients from Veracruz, MX and London, UK. Sequencing of 16S rRNA showed more consistency in microbial flora within IBS subtypes than between geographic location supporting the hypothesis that each subtype of IBS has a unique microbiome. Overall, the study is well conducted, and the findings will be of interest to the IBS and microbiome community. The central issue is that there are many prior publications in the role of microbiota in IBS subtypes that have heterogeneity or conflicting findings. Furthermore, most of the studies are phenomenological or correlative, with no clear mechanism(s) identified. I have a few suggestions that may improve the manuscript.

RESPONSE: We thank and value the time spent on our research work. It is our point of view, which we appreciate you noticed, that the contrasting findings found in the literature are due in part to the highly individualized microbial load in each subject, and its variation over time.

1.Is it feasible to show the overlap in the OTU for each IBS subtype between the MX and UK populations? As noted by the authors diet, lifestyle, and other factors (sex, medications, age et al.) can be confounding. However, are there any species/OTU that are “core” for each subtype? A Venn diagram (or enrichment analysis) may be helpful? I am not sure that the analysis at the phylum level is sufficient for resolving the key differences. Furthermore, as the authors note we don’t know what a “healthy” microbiome is for either population.

RESPONSE: The analysis at the phylum level is definitely not sufficient for resolving differences, which is why we also analyzed lower phylogenetic levels. The second suggestion actually came as a nice surprise because during manuscript preparation we did want to include a Venn Diagram but discarded the idea because it did not provide much new information (e.g. there were only a few OTUs that were unique to each subtype). We added this observation in the revised manuscript (in results, alpha diversity analyses).

2.As IBS presents as an activation of the immune system in the bowels, can the authors comment on the localization of any of the OTU identified? There are several tools like gutMgene (doi: 10.1093/nar/gkab786) that can be used, perhaps. As the authors note, the fecal microbiota represents a mix of microbes along the GI tract.

RESPONSE: We are particularly grateful with the reviewer for sharing such an interesting tool, but we are not sure what the reviewer means with “localization”. If the reviewer implied physical localization, this is not only impossible to determine but also biologically meaningless.

gutMgene contains manually curated relationships between microbes and metabolites, microbes and targets, and metabolites and targets, and has been cited mostly by researchers aiming to identify associations between specific genes and gut microbes (e.g. https://pubmed.ncbi.nlm.nih.gov/37328529/). The exploration of gutMgene led us to believe that its usefulness in studies like ours is limited, mainly because the nature of our data (we only have small regions of the huge pool of microbial 16S genes from fecal matter). Nonetheless, we explored the database for Streptoccus and noted that “The expression of several chemokine genes, some of which have reported antimicrobial properties, were negatively correlated with the relative abundance of Eubacterium rectale in the ileum, and Streptococcus and Eikenella in the rectum, suggesting that these species are the most susceptible to the activity of these chemokines”. We added this interesting observation in the discussion.

3.Can the authors include a 3x6 matrix (for anyone, e.g., Unifrac, Bray-Curtis) of the beta-diversity measures for 3 IBS subtypes and the two countries?

RESPONSE: During manuscript preparations, we tried multiple ways to display our results and this particular suggestion yielded a plot that is very difficult to read. However, please note our efforts in helping readers understand the implications of our results (e.g. with symbols [squares and circles] for each country).

Reviewer 2 Report

1Each group has fewer participants, and participants from the same country come from the same city. The 16S sequencing results are insufficient to represent the gut microbiota composition in IBS across the country. It is recommended to change the group name.

2The fecal samples are collected by the participants and stored in the refrigerator. Is there any human sampling error? How can we ensure the uniformity of sampling? Is the kit used for storing feces sealed, and will it be contaminated during sampling and storage?

3OUT and ASV are different sequence clusters, and the wording in the text needs to be more consistent.

4Some results do not have charts or figures(such as 3.2 and 3.4), and Figure S1-4 does not display the position in the main text.

5The article did not conduct statistical analysis on the duration of IBS illness among participants, and there may be differences in gut microbiota among different durations of illness. If relevant information is supplemented, it may have greater clinical significance.

6In result 3.1, it is stated that the proportion of IBS-D subtypes in the UK is higher compared with Mexico, and later, it is mentioned that the number and characteristics of IBS-D in the two cities are similar. It is recommended to change the wording, which may lead to ambiguity among readers.

7In the differential analysis of gut microbiota, the composition at the species level can explain the differences in specific gut microbiota between different groups. However, the article only analyzed and compared the differences in gut microbiota at the phylum and genus levels. Why was the gut microbiota not analyzed between different groups at the species level?

The expression of some sentences needs further modification and improvement.

Author Response

Dear reviewer,

Thank you for your time. Here you can find the response to your queries and suggestions. Also, please see the attachment to find our response to all reviewers.

*******

Reviewer 2

1、Each group has fewer participants, and participants from the same country come from the same city. The 16S sequencing results are insufficient to represent the gut microbiota composition in IBS across the country. It is recommended to change the group name.

RESPONSE: We agree with the reviewer in the potential confusion about the origin of the participants. However, we stated the names of both cities in Material and Methods and we simply cannot find any other group name that fits better.

2、The fecal samples are collected by the participants and stored in the refrigerator. Is there any human sampling error? How can we ensure the uniformity of sampling? Is the kit used for storing feces sealed, and will it be contaminated during sampling and storage?

RESPONSE: Those are indeed very important issues that are hard to control but we have no reason to believe that there was a contamination that would impact the microbial composition in each sample.

3、OTU and ASV are different sequence clusters, and the wording in the text needs to be more consistent.

RESPONSE: We agree and thank the reviewer for this observation. However, the word OTU was mentioned only once, to refer to the 16S database.

4、Some results do not have charts or figures (such as 3.2 and 3.4), and Figure S1-4 does not display the position in the main text.

RESPONSE: Section 3.2 is about sequence features and is better to describe with few words, and section 3.4 is about relative abundances, something that we mentioned only because that is the most commonly used approach. Supplementary figures S1-S4, and also S5-S6, are items that were included only to support supplementary sections and do not necessarily have to be displayed in the main text.

5、The article did not conduct statistical analysis on the duration of IBS illness among participants, and there may be differences in gut microbiota among different durations of illness. If relevant information is supplemented, it may have greater clinical significance.

RESPONSE: We understand and value this important observation. Indeed, the microbiome is always evolving and does change throughout time, but this was not part of the objectives of this research work.

6、In result 3.1, it is stated that the proportion of IBS-D subtypes in the UK is higher compared with Mexico, and later, it is mentioned that the number and characteristics of IBS-D in the two cities are similar. It is recommended to change the wording, which may lead to ambiguity among readers.

RESPONSE: In results section 3.1, we confirm the statement provided by the reviewer but noticed we used different wording (we said: “due to the high number of patients in the UK with IBS-D”). We rephrased those lines to make it easier to follow: “due to the high number of patients in the UK with IBS-D compared to the other subtypes”. With this change, we believe that the next sentence (“that the number and characteristics of IBS-D in the two cities are similar”) fits better.

7、In the differential analysis of gut microbiota, the composition at the species level can explain the differences in specific gut microbiota between different groups. However, the article only analyzed and compared the differences in gut microbiota at the phylum and genus levels. Why was the gut microbiota not analyzed between different groups at the species level?

RESPONSE: Excellent question. According to current knowledge on microbiology, the use of “species” does not hold much biological meaning (https://pubmed.ncbi.nlm.nih.gov/12142474/), in part because identical 16S sequences have been found in bacteria with highly divergent ecophysiologies (https://www.ncbi.nlm.nih.gov/pmc/articles/PMC492463/) and the huge differences within the same “species” (https://pubmed.ncbi.nlm.nih.gov/20623278/). That is why most researchers in gut microbial ecology tend to avoid the use and analysis of microbial populations at the “species” level. Please do note that we compared the data at the genus level to have a better idea of the potential differences among IBS subtypes.

Reviewer 3 Report

The paper seems very well done to me, and I don't see any particular issues. In my opinion, the main methodological limitation of the study is that it collected samples only once (we know that the microbiota can vary over time). Including healthy subjects in the study would probably have been useful. The text appears to be well-edited. On page 1, line 43, when discussing the influence of microbiota on gastrointestinal diseases, there are missing bibliographic references. I recommend including the following references: PMID: 37630649, PMID: 37317096 (studies that have analyzed the relationship between microbiota and disease, including IBS).

Author Response

Dear reviewer,

Thank you for your time. Here you can find the response to your queries and suggestions. Also, please see the attachment to find our response to all reviewers.

*******

Reviewer 3

The paper seems very well done to me, and I don't see any particular issues. In my opinion, the main methodological limitation of the study is that it collected samples only once (we know that the microbiota can vary over time). Including healthy subjects in the study would probably have been useful. The text appears to be well-edited. On page 1, line 43, when discussing the influence of microbiota on gastrointestinal diseases, there are missing bibliographic references. I recommend including the following references: PMID: 37630649, PMID: 37317096 (studies that have analyzed the relationship between microbiota and disease, including IBS).

RESPONSE: Thank you for the comments and suggestions. We revised the two suggested references, but the first focus on endometriosis and its possible connection with IBS, and the second is about gastroparesis. Please note our efforts in the original submission to include most of the main references related to IBS and microbiome, all throughout the manuscript.

Round 2

Reviewer 1 Report

The authors have adequately addressed my critiques.  I have no further concerns.